# Bile acids activate the antibacterial T6SS1 in the gut pathogen *Vibrio parahaemolyticus*

Sarah Schiffmann,[1] Shir Mass,[1] Dor Salomon[1]

**ABSTRACT**   The marine bacterium *Vibrio parahaemolyticus* is a major cause of seafood-borne gastroenteritis in humans and of acute hepatopancreatic necrosis disease in shrimp. Bile acids, produced by the host and modified into secondary bile acids by commensal bacteria in the gastrointestinal tract, induce the virulence factors leading to disease in humans and shrimp. Here, we show that secondary bile acids also activate this pathogen's type VI secretion system 1, a toxin delivery apparatus mediating interbacterial competition. This finding implies that *Vibrio parahaemolyticus* exploits secondary bile acids to activate its virulence factors and identify the presence of commensal bacteria that it needs to outcompete in order to colonize the host.

**IMPORTANCE**  Bacterial pathogens often manipulate their host and cause disease by secreting toxic proteins. However, to successfully colonize a host, they must also remove commensal bacteria that reside in it and may compete with them over resources. Here, we find that the same host-derived molecules that activate the secreted virulence toxins in a gut bacterial pathogen, *Vibrio parahaemolyticus*, also activate an antibacterial toxin delivery system that targets such commensal bacteria. These findings suggest that a pathogen can use one cue to launch a coordinated, *trans*-kingdom attack that enables it to colonize a host.

**KEYWORDS**   regulation, deoxycholate, secondary bile acid, type VI secretion system, *Vibrio parahaemolyticus*

*Vibrio parahaemolyticus* (*Vpara*) is a gram-negative bacterium common in marine and estuarine environments. It is a leading cause of seafood-borne gastroenteritis (1) and of acute hepatopancreatic necrosis disease (AHPND), which globally affects the shrimp farming industry (2). *Vpara* virulence in mammals is dependent on a toxin delivery apparatus named type III secretion system 2 (T3SS2) (3, 4), which is encoded within a horizontally acquired genomic island, VPaI-7 (5), in clinical isolates. AHPND in shrimp is caused by the PirAB binary toxin, encoded on a conjugative plasmid (6).

Although mutually exclusive in *Vpara* genomes (7), both T3SS2 and PirAB are induced by bile acids, an important component in the gastrointestinal tract of animals promoting lipid absorption, protein cleavage, and antimicrobial toxicity (8–11). Whereas bile-mediated activation of T3SS2 occurs via the VtrA/C receptor, which is also encoded within VPaI-7 (10, 11), AHPND-causing strains lack VtrA/C (7), and it is unclear how they sense bile acids to activate PirAB (9).

We previously reported that pathogenic *Vpara* strains employ a toxin delivery apparatus, named type VI secretion system 1 (T6SS1), to outcompete rival bacteria (12, 13). We reasoned that T6SS1 plays an indirect role during host infection by enabling *Vpara* to combat gut commensal bacteria and colonize the host. Notably, an antibacterial T6SS in the enteric pathogen *Citrobacter rodentium* was recently shown to promote the colonization of the host gut (14). Another gastrointestinal pathogen, *Salmonella*

Address correspondence to Dor Salomon, dorsalomon@mail.tau.ac.il.

The authors declare no conflict of interest.

See the funding table on p. 4.

Typhimurium, was shown to use the host bile acids as a cue to activate its antibacterial T6SS and establish a niche inside the host gut (15). Although environmental factors that regulate the *Vpara* T6SS1 have been investigated in several *Vpara* strains (12, 13), little is known about *Vpara* T6SS regulation by host factors. Nevertheless, a subset of *Vpara* T6SS1 genes was found to be transcriptionally induced during infection of an infant rabbit model (16), suggesting that conditions in the host gut can induce T6SS gene expression, even though whether specific host factors are involved remains unknown.

The abovementioned observations led us to hypothesize that *Vpara* uses bile sensing to induce its antibacterial T6SS1 in addition to its major virulence factors, thus gaining a competitive advantage during host infection. To test this hypothesis, we first set out to investigate the effect of bile acids on T6SS1 transcriptional regulation. Based on our previous delineation of the *Vpara* T6SS1 regulatory cascade in the type strain RIMD 2210633 (17), we constructed a reporter plasmid (pT6SS1[report]) in which the predicted promoter region of *vp1400*, the first gene in an operon encoding several conserved T6SS1 structural components, is fused to a promoterless *cat* and *gfp* reporter cassette. To verify that this reporter reliably represents T6SS1, we monitored GFP expression in *Vpara* strain RIMD 2210633 under conditions known to induce or repress T6SS1, in defined M9 media containing 3% (wt/vol) NaCl at 30°C. As expected, GFP fluorescence was detected when *hns*, encoding a histone-like nucleoid structuring protein previously shown to repress T6SS1 (18), was deleted (Fig. 1A). Notably, GFP expression was higher at earlier time points and declined until reaching a steady state after ~120 minutes. We hypothesize that this is caused by quorum-sensing signaling affected by the change in cell density during the incubation period. Indeed, we previously showed that the quorum-sensing master regulator, OpaR, negatively affects the expression of T6SS1 (12, 19). Fluorescence was also detected upon the addition of phenamil, an inhibitor of the polar flagella motor shown to induce T6SS1 by mimicking surface-sensing activation (12), to the media of wild-type *Vpara* (Fig. 1A). In contrast, phenamil addition did not induce GFP fluorescence in a strain in which we deleted the T6SS1-positive regulator, *tfoY* (19). Similar results were obtained when we monitored the expression of the *cat* reporter gene, providing chloramphenicol resistance, which manifested as growth on agar plates containing chloramphenicol (Fig. 1B). Therefore, we conclude that pT6SS1[report] reliably represents the status of T6SS1 activation.

To investigate the effect of bile acids on T6SS1 transcription, we chose two secondary bile acids, which are bile acids modified in the host gut by commensal bacteria (8): taurodeoxycholate (TDC) and deoxycholate (DOC), which are strong and intermediate inducers of the *Vpara* T3SS2, respectively (20). Remarkably, the addition of DOC to the growth media at a concentration previously shown to activate T3SS2 [0.025% (wt/vol)] (20) resulted in the induction of GFP fluorescence from the T6SS1 reporter to levels comparable to those of the known inducer, phenamil (Fig. 1C). Surprisingly, TDC, which is a strong activator of T3SS2, had only a mild effect on the T6SS1 reporter (Fig. 1C). Neither bile acids hampered *Vpara* growth at these concentrations (Fig. S1); however, a high concentration of DOC [0.05% (wt/vol)] appears to have stimulated bacterial growth, for reasons that remain to be investigated, without further inducing GFP fluorescence. Moreover, neither DOC nor phenamil induced expression from the promoter of a T6SS1-unrelated gene, *vpa1270*, thus indicating that the observed phenotypes do not result from global effects on *Vpara* transcription or translation (Fig. S2). Taken together, these results suggest that bile acids induce T6SS1.

Next, we sought to determine whether the observed bile acid-induced T6SS1 transcriptional activation translates to the assembly of a functional T6SS1 apparatus. To this end, we investigated the effect of DOC on T6SS1 activity by monitoring the expression and secretion of the hallmark T6SS structural secreted protein, VgrG1 (21). In agreement with the results obtained for the transcriptional reporter, DOC induced VgrG1 expression and secretion to a level comparable to that of phenamil (Fig. 1D). Moreover, intracellular accumulation of VgrG1 was evident upon inactivation of T6SS1 by deleting the conserved structural component Hcp1 (Δ*hcp1*) (22). Notably, although inactivation of

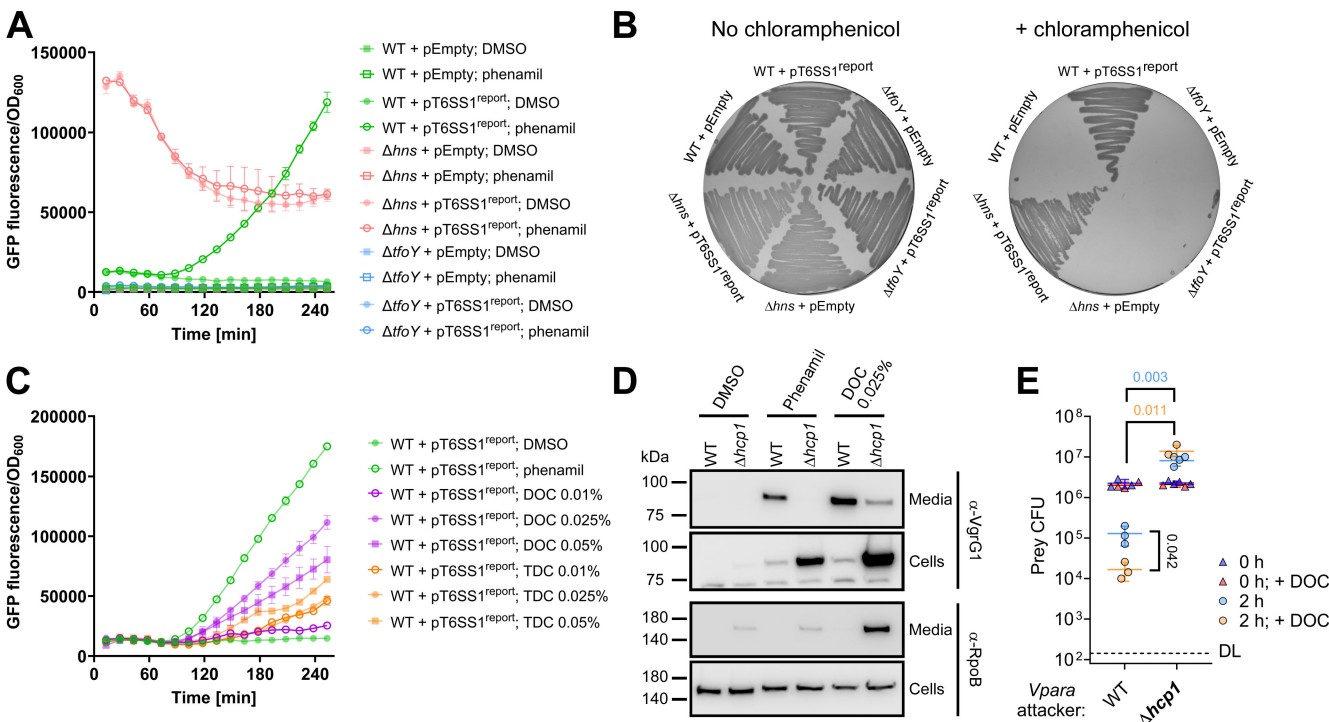

**FIG 1** Bile acids activate *Vibrio parahaemolyticus* T6SS1. (A and C) Fluorescence intensity over time in the indicated *V. parahaemolyticus* RIMD 2210633 strains carrying an empty plasmid (pEmpty) or a plasmid containing a *gfp* and *cat* reporter cassette fused to the promoter of *vp1400* (pT6SS1report), measured as GFP fluorescence to $OD_{600}$ readings (arbitrary units). Bacteria were grown at 30°C in M9 media supplemented with 3% (wt/vol) NaCl and kanamycin (250 μg/mL) to maintain the plasmids. Where indicated, the media were supplemented with phenamil (20 μM), DMSO [20% (vol/vol), added to the media as a control at the same volume of the phenamil solution], or the indicated concentrations of deoxycholate (DOC) or taurodeoxycholate (TDC) (wt/vol). Data are shown as the mean ± SD, $n = 3$ independent replicates. (B) Growth of the indicated *V. parahaemolyticus* RIMD 2210633 strains at 30°C on MLB [LB with 3% (wt/vol) NaCl] agar plates containing kanamycin (250 μg/mL) to maintain the plasmids, with or without chloramphenicol (10 μg/mL) to monitor *cat* expression. (D) Expression (cells) and secretion (media) of VgrG1 from the indicated *V. parahaemolyticus* RIMD 2210633 strains grown for 4 h in M9 media supplemented with 3% (wt/vol) NaCl and either phenamil, DMSO (as detailed for A,C), or DOC at 30°C. RNA polymerase β subunit (RpoB) was used as a loading and lysis control. (E) Viability counts (colony-forming units) of *Vibrio natriegens* prey strains before (0 h) and after (2 h) co-incubation with the indicated *Vpara* attacker strain, either pre-incubated with 0.025% (wt/vol) DOC or not, on MLB agar plates at 30°C. The statistical significance between samples at the 2-h time point was calculated using an unpaired, two-tailed Student's *t* test. Data are shown as the mean ± SD; $n = 3$ independent competition replicates. In A–E, results from a representative experiment out of at least three independent experiments are shown. WT, wild type.

T6SS1 hampered the DOC-induced secretion of VgrG1 compared to the wild-type strain, a low level of VgrG1 signal was detected in the "media" fraction of the DOC-treated Δ*hcp1* strain; this observation can be explained by cell lysis apparent in the Δ*hcp1* strains (see the α-RpoB panel in Fig. 1D), which was exacerbated in the presence of DOC. These results demonstrate that bile acids activate *Vpara* T6SS1.

Lastly, to confirm that bile acid-induced T6SS1 is biologically functional, we investigated whether DOC can potentiate the T6SS1-mediated antibacterial activity and provide *Vpara* with an advantage during inter-bacterial competition. As shown in Fig. 1E, we found that pre-incubation of *Vpara* attacker strains with DOC enabled them to kill 10-fold more *Vibrio natriegens* prey bacteria during competition, compared with attackers not treated with DOC, in a T6SS1-dependent manner. These results confirm that bile acid-induced T6SS1 is biologically functional and that DOC enhances to competitive fitness of *Vpara*.

## CONCLUSIONS

Our results suggest that *Vpara* use host-derived molecules, indicative of the presence of commensal bacteria, as a signal that activates virulence factors and antibacterial

determinants playing a role in interbacterial competition. Therefore, bile acids may be exploited by *Vpara* to establish a niche inside the host. Future *in vivo* competition assays using animal models are required to test this prediction. Notably, the bile acid receptor that regulates T6SS1 remains to be identified. Moreover, secondary bile acids were shown to repress the T6SS in another gut pathogen, *V. cholerae* (23), indicating that exploiting commensal bacteria-made molecules to activate antibacterial determinants in the host gut may not be a universal bacterial strategy.

## ACKNOWLEDGMENTS

We thank members of the Salomon lab for helpful discussions. This study was partly funded by a research grant awarded by the Tel Aviv University Center for Combatting Pandemics and by the Israel Science Foundation (grant number 1362/21) to D.S.

## AUTHOR AFFILIATION

[1]Department of Clinical Microbiology and Immunology, School of Medicine, Faculty of Medical and Health Sciences, Tel Aviv University, Tel Aviv, Israel

## AUTHOR ORCIDs

Dor Salomon  http://orcid.org/0000-0002-2009-9453

## FUNDING

| Funder | Grant(s) | Author(s) |
| --- | --- | --- |
| Israel Science Foundation | 1362/21 | Dor Salomon |

## AUTHOR CONTRIBUTIONS

Sarah Schiffmann, Formal analysis, Investigation, Methodology, Writing – review and editing | Shir Mass, Investigation, Methodology, Writing – review and editing | Dor Salomon, Conceptualization, Formal analysis, Funding acquisition, Supervision, Writing – original draft

## ADDITIONAL FILES

The following material is available online.

### Supplemental Material

**Supplemental materials (Spectrum01181-24-S0001.docx).** Supplemental methods, figures, tables, and references.

### Open Peer Review

**PEER REVIEW HISTORY (review-history.pdf).** An accounting of the reviewer comments and feedback.

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
