## [Reviewer comments · Microbiology Spectrum]

Microbiology Spectrum

Bile acids activate the antibacterial T6SS1 in the gut pathogen *Vibrio parahaemolyticus*

Sarah Schiffmann, Shir Mass, and Dor Salomon

Corresponding Author(s): Dor Salomon, Tel Aviv University

Review Timeline:

Submission Date:	May 12, 2024
Editorial Decision:	June 8, 2024
Revision Received:	June 20, 2024
Accepted:	July 2, 2024

Editor: Carlos Blondel

Reviewer(s): The reviewers have opted to remain anonymous.

Transaction Report:

DOI: <https://doi.org/10.1128/spectrum.01181-24>

Re: Spectrum01181-24 (Bile acids activate the antibacterial T6SS1 in the gut pathogen *Vibrio parahaemolyticus*)

Dear Prof. Dor Salomon:

Thank you for the privilege of reviewing your work. Below you will find my comments, instructions from the Spectrum editorial office, and the reviewer comments.

Revision Guidelines

Sincerely,
Carlos Blondel
Editor
Microbiology Spectrum

Reviewer #1 (Comments for the Author):

This is an interesting observation study that suggests that the T6SS1 in Vp is expressed when the bacteria are exposed to bile acids. I only have minor edits and recommendations.

1. line 54-55. It is recommended to state that conditions in the host gut can induce T6SS gene expression. It is not clear whether specific host factors are involved (at least not yet).

2. *Citrobacter* T6SS has been shown to be required for host intestinal colonization in direct competition with mouse intestinal flora. It would be appropriate to cite that paper in the introduction (DOI: 10.1016/j.celrep.2022.110731)
3. DOC at 0.05% seems to stimulate growth. Also the gene expression does not seem dose-dependent above 0.025%, so something physiological could be happening. The authors should comment on this as it was not observed for taurodeoxycholate.
4. The authors are strongly encouraged to explore whether the gene expression they are observing is biologically functional. It is unclear why a competition killing assay has not been done in the presence of bile salts. The 2013 PLoSone paper from this group demonstrates 4 hr killing assays (240 min), which in this study shows T6SS gene expression. A simple killing assay would greatly enhance this study. It is puzzling that such an experiment is not presented in the manuscript.

Reviewer #2 (Comments for the Author):

This work by Schiffmann et al describes an interesting result that will be of value to the *Vibrio parahaemolyticus* community. They show that bile salt DOA stimulated the T6SS that could impact ability to occupy niche or compete with microbiota during infection of humans or shrimps. The paper is a single observation consistent with the observation format. I would like the authors only to add comments on:

- 1) why is the GFP fluorescence is high immediately following dilution of the Δ hns mutant and then goes down. I think this is likely known from prior work that it accumulates in stationary phase. The finding is not central to any finding in the paper so no additional experiments, just a comment of explanation.
- 2) Is there an assay pre-existing for T6SS killing of for example *E coli* in vitro? would it be sufficiently negative at baseline that it could be shown that T6SS killing function is stimulated by adding DOC to the agar. This would provide biological impact to this nice reporter observations given. However, it may be out of scope or not possible given the current state of Vp research or the need to engineer strains so in lieu of adding experiment, some speculation in text will suffice.

Point-by-point reply to reviewers' comments

Reviewer #1 (Comments for the Author):

This is an interesting observation study that suggests that the T6SS1 in *Vp* is expressed when the bacteria are exposed to bile acids. I only have minor edits and recommendations.

We thank the reviewer for these kind words.

1. line 54-55. It is recommended to state that conditions in the host gut can induce T6SS gene expression. It is not clear whether specific host factors are involved (at least not yet).

The sentence was rephrased to read: "Nevertheless, a subset of *Vpara* T6SS1 genes was found to be transcriptionally induced during infection of an infant rabbit model, suggesting that conditions in the host gut can induce T6SS gene expression, even though whether specific host factors are involved remains unknown".

2. *Citrobacter* T6SS has been shown to be required for host intestinal colonization in direct competition with mouse intestinal flora. It would be appropriate to cite that paper in the introduction (DOI: 10.1016/j.celrep.2022.110731)

We thank the reviewer for this suggestion. We now cite this important report: "Notably, an antibacterial T6SS in the enteric pathogen *Citrobacter rodentium* was recently shown to promote the colonization of the host gut".

3. DOC at 0.05% seems to stimulate growth. Also the gene expression does not seem dose-dependent above 0.025%, so something physiological could be happening. The authors should comment on this as it was not observed for taurodeoxycholate.

We thank the reviewer for pointing this out. We now mention this observation in the text, although, at this time, we do not know the reason behind it: "Neither bile acid hampered *Vpara* growth at these concentrations (**Fig. S1**); however, a high concentration of DOC (0.05% [wt/vol]) appears to have stimulated bacterial growth, for reasons that remain to be investigated, without further inducing GFP fluorescence".

4. The authors are strongly encouraged to explore whether the gene expression they are observing is biologically functional. It is unclear why a competition killing assay has not been done in the presence of bile salts. The 2013 PloSone paper from this group demonstrates 4 hr killing assays (240 min), which in this study shows T6SS gene expression. A simple killing assay would greatly enhance this study. It is puzzling that such an experiment is not presented in the manuscript.

We thank the reviewer for this suggestion. We note that a simple competition assay, such as the one previously reported by our group, is not appropriate to test the effect of bile acids. The

reason for this is that surface sensing, which is activated once the bacteria are spotted on the agar plates for competition, already induces the T6SS1 in *Vpara*, thereby masking any effect that bile acids may have.

Nevertheless, we hypothesized that bile acids may potentiate the T6SS1-mediated killing activity of *Vpara* if the bacteria are pre-incubated with them. Therefore, to unmask the potential activation of the T6SS1 by bile acids, we devised an alternative competition assay: first, we pre-incubated the *Vpara* attacker strains with bile acids for two hours to induce T6SS1 activation. Next, we washed the cells and mixed them with the prey bacteria, and then we spotted these mixtures on the agar plates for a two-hour-long competition. As shown in the new **Fig. 1E**, we found that *Vpara* attackers pre-incubated with bile acids kill tenfold more prey bacteria than attackers not treated with bile acids. These results confirm that bile acid-induced T6SS1 is biologically functional and able to enhance the competitive fitness of *Vpara*.

Reviewer #2 (Comments for the Author):

This work by Schiffmann et al describes an interesting result that will be of value to the *Vibrio parahaemolyticus* community. They show that bile salt DOA stimulated the T6SS that could impact ability to occupy niche or compete with microbiota during infection of humans or shrimps. The paper is a single observation consistent with the observation format.

We thank the reviewer for these kind words.

I would like the authors only to add comments on:

1) why is the GFP fluorescence is high immediately following dilution of the Δ hns mutant and then goes down. I think this is likely known from prior work that it accumulates in stationary phase. The finding is not central to any finding in the paper so no additional experiments, just a comment of explanation.

We thank the reviewer for mentioning this observation. We have added a possible explanation to this phenomenon, which we think is connected to quorum-sensing: "Notably, GFP expression was higher at earlier timepoints and declined until reaching a steady state after ~120 minutes. We hypothesize that this is caused by quorum-sensing signaling affected by the change in cell density during the incubation period. Indeed, we previously showed that the quorum-sensing master regulator, OpaR, negatively affects the expression of T6SS1".

2) Is there an assay pre-existing for T6SS killing of for example *E coli* in vitro? would it be sufficiently negative at baseline that it could be shown that T6SS killing function is stimulated by adding DOC to the agar. This would provide biological impact to this nice reporter observations given. However, it may be out of scope or not possible given the current state of *Vp* research or the need to engineer strains so in lieu of adding experiment, some speculation in text will suffice.

We thank the reviewer for this suggestion. We note that a simple competition assay, such as the one previously reported by our group, is not appropriate to test the effect of bile acids. The reason for this is that surface sensing, which is activated once the bacteria are spotted on the agar plates for competition, already induces the T6SS1 in *Vpara*, thereby masking any effect that bile acids may have.

Nevertheless, we hypothesized that bile acids may potentiate the T6SS1-mediated killing activity of *Vpara* if the bacteria are pre-incubated with them. Therefore, to unmask the potential activation of the T6SS1 by bile acids, we devised an alternative competition assay: first, we pre-incubated the *Vpara* attacker strains with bile acids for two hours to induce T6SS1 activation. Next, we washed the cells and mixed them with the prey bacteria, and then we spotted these mixtures on the agar plates for a two-hour-long competition. As shown in the new **Fig. 1E**, we found that *Vpara* attackers pre-incubated with bile acids kill tenfold more prey bacteria than attackers not treated with bile acids. These results confirm that bile acid-induced T6SS1 is biologically functional and able to enhance the competitive fitness of *Vpara*.

Re: Spectrum01181-24R1 (Bile acids activate the antibacterial T6SS1 in the gut pathogen *Vibrio parahaemolyticus*)

Dear Prof. Dor Salomon:

Your manuscript has been accepted, and I am forwarding it to the ASM production staff for publication. Your paper will first be checked to make sure all elements meet the technical requirements. ASM staff will contact you if anything needs to be revised before copyediting and production can begin. Otherwise, you will be notified when your proofs are ready to be viewed.

Sincerely,
Carlos Blondel
Editor
Microbiology Spectrum